# Bi-EB: Empirical Bayesian Biclustering for Multi-Omics Data Integration Pattern Identification among Species

**DOI:** 10.3390/genes13111982

**Published:** 2022-10-30

**Authors:** Aida Yazdanparast, Lang Li, Chi Zhang, Lijun Cheng

**Affiliations:** 1Center for Computational Biology and Bioinformatics, School of Medicine, Indiana University, Indianapolis, IN 46202, USA; 2Department of Bio-Health Informatics, School of Informatics, Indiana University, Indianapolis, IN 46202, USA; 3Department of Medical and Molecular Genetics, School of Medicine, Indiana University, Indianapolis, IN 46202, USA; 4Department of Biomedical Informatics, College of Medicine, Ohio State University, Columbus, OH 43210, USA

**Keywords:** biclustering, multi-omics data analysis, breast cancer, tumor and cancer cell lines

## Abstract

Although several biclustering algorithms have been studied, few are used for cross-pattern identification across species using multi-omics data mining. A fast empirical Bayesian biclustering (Bi-EB) algorithm is developed to detect the patterns shared from both integrated omics data and between species. The Bi-EB algorithm addresses the clinical critical translational question using the bioinformatics strategy, which addresses how modules of genotype variation associated with phenotype from cancer cell screening data can be identified and how these findings can be directly translated to a cancer patient subpopulation. Empirical Bayesian probabilistic interpretation and ratio strategy are proposed in Bi-EB for the first time to detect the pairwise regulation patterns among species and variations in multiple omics on a gene level, such as proteins and mRNA. An expectation–maximization (EM) optimal algorithm is used to extract the foreground co-current variations out of its background noise data by adjusting parameters with bicluster membership probability threshold *Ac*; and the bicluster average probability *p*. Three simulation experiments and two real biology mRNA and protein data analyses conducted on the well-known *Cancer Genomics Atlas* (TCGA) and *The Cancer Cell Line Encyclopedia* (CCLE) verify that the proposed Bi-EB algorithm can significantly improve the clustering recovery and relevance accuracy, outperforming the other seven biclustering methods—Cheng and Church (CC), xMOTIFs, BiMax, Plaid, Spectral, FABIA, and QUBIC—with a recovery score of 0.98 and a relevance score of 0.99. At the same time, the Bi-EB algorithm is used to determine shared the causality patterns of mRNA to the protein between patients and cancer cells in TCGA and CCLE breast cancer. The clinically well-known treatment target protein module estrogen receptor (ER), ER (p118), AR, BCL2, cyclin E1, and IGFBP2 are identified in accordance with their mRNA expression variations in the luminal-like subtype. Ten genes, including CCNB1, CDH1, KDR, RAB25, PRKCA, etc., found which can maintain the high accordance of mRNA–protein for both breast cancer patients and cell lines in basal-like subtypes for the first time. Bi-EB provides a useful biclustering analysis tool to discover the cross patterns hidden both in multiple data matrixes (omics) and species. The implementation of the Bi-EB method in the clinical setting will have a direct impact on administrating translational research based on the cancer cell screening guidance.

## 1. Introduction

Co-regulated gene module detection will assist us in identifying its biological functions or molecular pathways. Conventional clustering methods uncover co-expressed genomic profiles across all samples; however, they cannot detect shared patterns in a subset of genes and among a subset of samples, called co-clusters or biclusters[1]. The biclustering algorithm, first introduced in 2000 by Cheng and Church (CC) [2], was designed to discover gene modules among a subset of samples. So far, a number of probabilistic model-based biclustering algorithms focused on finding biclusters [3] that characterize the hierarchical signal and noise structure of the data, such as the plaid model introduced by Lazzeroni and Owen [4]. The plaid model is composed of several layers of biclusters, and each bicluster is decided by column means (samples) and row means (genes). The bicluster search algorithm was based on an iterative fitting procedure. In the plaid model, the error terms were assumed to follow the normal distribution. Because of the hierarchical structure of the biclustering model in the plaid model, Bayesian models based on the normal distribution assumption with conjugated priors were then developed [5,6] to update biclustering with more accuracy. Comparing to the plaid model, these Bayesian models have a clear theoretical advantage when posterior probability is used. The ability to consider model uncertainty within a single framework towards frequentist techniques for justification is important. For example, the recent work by Amar et al. [7] expanded the Bayesian biclustering model that handles categorical data. Kirk et al. [8] extended the Bayesian clustering that integrated several different datasets, but it was not a biclustering algorithm [9].

In these empirical Bayesian models, Gibbs sampling schemes were developed and implemented to estimate the model parameters and detect biclusters for the underlying probabilistic distribution inner data. The empirical Bayes mixture model is a valuable alternative approach for Bayesian models. Computationally, its expectation–maximization (EM) algorithm is usually less expensive than Bayesian model’s Gibb sampling approaches. Chekouo and Murua (2015) [9] recently formulated the plaid model into an empirical Bayes mixture model to detect biclusters using the EM algorithm. In this model, both the biclusters and the background data were assumed to share the same variance, while the sample mean, and gene mean expressions were assumed as fixed effects in each bicluster. 

As compared to all the existing clustering algorithms, in this paper, we made four major innovative contributions to the empirical Bayes mixture model to detect biclusters. Firstly, our model provides a different variance structure between the biclusters and the background. Secondly, our model assumes that sample means and row means follow normal distributions too. In other words, we use random-effect model formulation instead of fixed effects. Thirdly, for the first time, we provide comprehensive EM algorithm derivation for the biclutering mixture model. Fourthly, we recognize that the EM algorithm itself only provides the probabilistic estimates for the biclustering memberships, but it does not really group the data into biclusters. We further develop an algorithm that automatically searches and groups data into biclusters based on the probabilities estimated after the EM algorithm.

Large-scale omics profiling has been conducted to investigate the molecular signatures of diseases. Using fast-evolving high-throughput technologies, including transcriptome, DNA copy number alterations, and proteomic data, can provide us with tremendous opportunities to examine disease-specific biological pathways and molecular functions. For example, *The Cancer Genomics Atlas* (TCGA) [10] and *The Cancer Cell Line Encyclopedia* (CCLE) [11] are exemplified omics profiling projects for human cancer tumor samples and cancer cell lines. Cell lines have the advantages of being easily grown in the in vitro experiment system, cost-effective, and amenable to the high throughput testing of therapeutic agents. Data integration between cell lines and tumors can translate molecular features from cell culture models to cancer patients [12,13], and the goal is to build predictive key signatures for molecular mechanism detection and drug targets. Characterizing key molecular alterations in both patient samples and cell lines and discovering therapeutic targets are some of the primary goals in precision cancer medicine.

Cancer subtype stratification has become a critical component of disease characterization. Research efforts have focused on how the classification of these subtypes could provide information on influence treatment planning [14]. Clustering methods are the most common pattern recognition approach in classifying cancer subtypes. With regards to breast cancer, as the example used in this paper, the major classification schemes are based on mRNA expression profiling which are often referred to as intrinsic subtypes in breast cancer, which include: luminal A, luminal B, HER2-enriched, basal-like, and normal breast-like [13,14,15]. Further clustering analysis on the triple-negative breast cancer transcriptome revealed additional triple-negative breast cancer (TNBC) six subtypes: basal-like 1 (BL1), basal-like 2 (BL2), immune-modulatory (IM), mesenchymal-like (M), mesenchymal stem-like (MSL), and luminal androgen receptor (LAR) [16]. Recently, using addition histopathological data, IM and MSL TNBC subtypes have been recognized as they helped to infiltrate lymphocytes and tumor-associated stromal cells. Hence, only four TNBC subtypes were confirmed: BL1, BL2, M, and LAR [17].

Tumor-derived cell lines have long been used to study the underlying biologic processes in cancer, as well as screening platforms for discovering and evaluating the efficacy of anticancer therapeutics. Proper cell models for cancer primary tumors have long been the focal point in cancer-based research [13]. The identification of key common gene modules (clustering) in an in vitro model by using large number of cancer cell models and tumors is a promising approach for the development of targeted treatments. Previous studies clustered cell line and tumor samples separately with the goal of successfully identifying several major cancer subtypes, and their associated molecular signatures in both data after clustering [18,19]. However, there has not been any attempt to find transcriptome signatures or patterns that are mutually shared between the cell line and tumor by clustering them simultaneously. Secondly, clustering is applied to all samples and all transcriptomes. However, if not all the genes share the similarly among not all the samples, a biclustering method shall be considered a more suitable approach. We hypothesize that a subset of patient samples and cell lines shares the molecular signatures in gene subsets, though not all genes or all samples. This hypothesis cannot be answered by the traditional clustering methods, and a biclustering method is indeed an ideal solution. Thirdly, there has not yet been any effort to integrate protein and transcriptome data together in clustering breast cancer samples. 

In this paper, an empirical Bayesian biclustering (Bi-EB) algorithm is proposed to identify translational gene sets shared between cancer cell lines and primary tumors based on mRNA and proteomic data or copy number variations (CNVs) and mRNA data. An EM algorithm is developed to conduct estimation and inference in the bicluster analyses. Our bicluster searching starts from a seed, such as a druggable target gene, and it detects interesting gene modules shared between cancer cell lines versus patient tumor samples. Using Bi-EB, gene modules of mRNA and proteomics are explored in two breast cancer subtypes: luminal A/B and basal-like subtypes.

The article is organized as follows. In the Results section, Bi-EB is used to search for shared gene modules between patient tumor versus breast cancer cell line samples in datasets TCGA and CCLE. In Section 2, we present the Bi-EB model. In Section 3, we compare the Bi-EB algorithm to the other biclustering methods in the simulated data. In Section 4, our proposed Bi-EB algorithm is further discussed.

## 2. Materials and Methods

### 2.1. Materials

Emerging next-generation sequencing (NGS) and microarray techniques, as well as large-scale cancer screening data, can help to achieve this goal. Databases such as the database CCLE (http://www.broadinstitute.org/ccle, accessed on 2 February 2022) [11] provides public access to genomic data over 1000 cancer cell lines by RNA sequencing (RNA-seq; 1019 cell lines), whole-exome sequencing (326 cell lines), whole-genome sequencing (329 cell lines), and reverse-phase protein array (RPPA; 899 cell lines). The TCGA (http://cancergenome.nih.gov/, accessed on 1 March 2022) [10] project (https://cancergenome.nih.gov/abouttcga/overview, accessed on 1 March 2022) has now provided detailed molecular compositions for over 11,000 cancer patients’ whole-genome sequencing, RNA-seq, and RPPA data from at least 33 anatomic sites. The RPPA and mRNA expression, copy number, and mutation profiles from TCGA and CCLE are used to calculate similarities between tumors and cell lines. To simulate breast cancer, these subtypes and data refer to literature datasets [10,11]. Then, the ideal cell lines for cancer experiments are found by combining the results with gene ontology functional similarity. All data are provided in Appendix A. All mRNA expression data are normalized as reads per kilo base of transcript per million mapped reads (RPKM) first. Then, these gene and protein expression profiles are normalized in the literature [20,21] models for a Z-score, which is used in the further mRNA–protein ratio calculation.

### 2.2. Methods

#### The Empirical Bayes Biclustering (Bi-EB) Model

The Bi-EB model is used to identify co-regulated biclusters across tumors and cancer cells. Figure 1 shows the principle of empirical Bayes biclustering model (Bi-EB). The Bi-EB model assumes that the data follow both the background model and the bicluster model. Let us assume that we have *I* genes (*rows*) and *J* samples (*columns*). It consists of the *grand mean*; the between-gene (*row) variation*; the between-sample (*column) variation*; and *noise* from the background and bicluster, respectively. Please note that the sample set contains both primary tumors and cancer cell lines. Denote $Y=\{y_{ij}\}$ as the data matrix. *Y* can either be the transcriptome, proteome, or their ratios. We assume that *y_ij_* follows a mixture model.
(1)$$\;y_{ij}=\left\{\begin{array}{c} \mu_{1}+\alpha_{1i}+\beta_{1j}+\varepsilon_{1ij}\;\;\;\;\;\;\;\;\;if\;y_{ij}\;\in B\;\\ \mu_{2}+\alpha_{2i}+\beta_{2j}+\varepsilon_{2ij}\;\;\;\;\;\;\;\;\;if\;y_{ij}\;\notin B \end{array}\right.$$

In model (1), $\mu_{1}$ is the grand mean of data in the bicluster ***B***. The between-gene variation$\;\alpha_{1i}∼N\left(0,\delta_{1}^{2}\right)$, the sample variation$\;\beta_{1j}∼N\left(0,\tau_{1}^{2}\right)$, and the overall noise in bicluster$\;\varepsilon_{1ij}∼N\left(0,\sigma_{1}^{2}\right)$. On the other hand, $\mu_{2}$ is the grand mean of the background, and the between-gene variation, sample variation, and overall noise are$\;\alpha_{2i}∼N\left(0,\delta_{2}^{2}\right)$, $\beta_{2j}\;\sim \;N\left(0,\tau_{2}^{2}\right)$, and $\varepsilon_{2ij}∼N\left(0,\sigma_{2}^{2}\right)$, respectively. These biclusters are taken from data cells, whereby $\mu_{1}$ of the bicluster is the difference from $\mu_{2}$ of the background. Let $z=\left\{z_{1i},z_{2j}\right\}$ be a Bernoulli random variable, $z_{1i}\sim B\left(n_{1},p_{1}\right),z_{2j}\;\sim B\left(n_{2},p_{2}\right)$, which denotes the data point’s membership in the bicluster. If $z_{1i}=1$ and $z_{2j}=1$, $y_{ij}$ belongs to the bicluster ***B*** ($y_{ij}\in B$), and *if* $z_{1i}=0$ or $z_{2j}=0$, $y_{ij}$ belongs to the background model, as shown in Figure 1b. For the sake of simplicity, let $\theta =\left\{\theta_{1},\theta_{2}\right\}=\left\{\left\{\mu_{1},\alpha_{1i},\beta_{1j},\sigma_{1}\right\},\left\{\mu_{2},\alpha_{2i},\beta_{2j},\sigma_{2}\;\right\}\right\}.$ The complete joint likelihood function for $\left(Y,Z;\theta \right)\;$is defined as following in a threshold Ac:(2)$$L_{Ac}=\overset{I}{\underset{i=1}{\prod }}\overset{J}{\underset{j=1}{\prod }}Pr{\left(y_{ij}|\theta_{1}\right)}^{z_{1i}\times z_{2j}}Pr{\left(y_{ij}|\theta_{2}\right)}^{\left(1-z_{1i}\times z_{2j}\right)}Pr\left(z_{1i}\right)Pr\left(z_{2j}\right)$$
where$\;\mathrm{Pr}\left(y_{ij}|\theta_{k}\right)\;$follows the Gaussian distribution (3).
(3)$$\mathrm{Pr}\left(y_{ij}|\theta_{k}\right)=\frac{1}{\sqrt{2\pi }\sigma_{k}\delta_{k}^{}\tau_{k}^{}}exp\frac{-{\left(y_{ij}-\mu_{k}-\alpha_{ki}-\beta_{kj}\right)}^{2}}{2\sigma_{k}^{2}}exp\frac{-\alpha_{ki}{}^{2}}{2\delta_{k}^{2}}exp\frac{-\beta_{kj}{}^{2}}{2\tau_{k}^{2}}\;\mathrm{Pr}\left(z_{1i}\right)=p_{1}^{z_{1i}}{\left(1-p_{1}\right)}^{1-z_{1i}},\;\mathrm{Pr}\left(z_{2j}\right)=p_{2}^{z_{2j}}{\left(1-p_{2}\right)}^{1-z_{2j}}$$


**The EM Algorithm**


In the expectation–maximization (EM) algorithm, the **E step** is an iterative marginal distribution used to find the (local) maximum likelihood (Equation (7)) in the assignment of genes and arrays to biclusters using Formulas (4)–(6). The **M step** is used to look for the optimal parameters of the model using Formula (8) until the difference of (local) maximum likelihood reaches the threshold in iterations *t* and *t*+1 (Formula (9)). The **EM** algorithm follows four steps: 

**(i) Starting Values:**$P^{\left(0\right)}=\left(p_{1}^{\left(0\right)},\;p_{2}^{\left(0\right)}\right)$ is set at iteration t = 0. The parameters$\;\theta^{\left(0\right)}=\left(\theta^{\left(0\right)}{}_{1},\theta^{\left(0\right)}{}_{2}\right)$ are calculated based on $P^{\left(0\right)}$.

**(ii) E-Step:**$z_{1i}^{\left(t\right)}=E\left(z_{1i}|\theta^{\left(t\right)},p_{1}^{\left(t\right)},\;p_{2}^{\left(t\right)},\;Y\right)$ and $z_{2j}^{\left(t\right)}=E\left(z_{2j}|\theta^{\left(t\right)},p_{1}^{\left(t\right)},\;p_{2}^{\left(t\right)},\;Y\right)$ are both calculated.
(4)$$z_{1i}\sim p_{1}^{z_{1i}}{\left(1-p_{1}\right)}^{1-z_{1i}}\overset{J}{\underset{j=1}{\prod }}\mathrm{Pr}{\left(y_{ij}|\theta_{1}\right)}^{z_{1i}\times z_{2j}}\mathrm{Pr}{\left(y_{ij}|\theta_{2}\right)}^{\left(1-z_{1i}\times z_{2j}\right)}p_{2}^{z_{2j}}{\left(1-p_{2}\right)}^{1-z_{2j}}\;,\;\mathrm{and}\;z_{1i}^{\left(t\right)}=\int \left\{z_{1i}p_{1}^{z_{1i}}{\left(1-p_{1}\right)}^{1-z_{1i}}\times \overset{J}{\underset{j=1}{\prod }}\int \mathrm{Pr}{\left(y_{ij}|\theta_{1}\right)}^{z_{1i}\times z_{2j}}\mathrm{Pr}{\left(y_{ij}|\theta_{2}\right)}^{\left(1-z_{1i}\times z_{2j}\right)}p_{2}^{z_{2j}}{\left(1-p_{2}\right)}^{1-z_{2j}}dz_{2j}\right\}dz_{1i}$$
(5)$$z_{2j}\sim p_{2}^{z_{2j}}{\left(1-p_{2}\right)}^{1-z_{2j}}\overset{I}{\underset{i=1}{\prod }}\mathrm{Pr}{\left(y_{ij}|\theta_{1}\right)}^{z_{1i}\times z_{2j}}\mathrm{Pr}{\left(y_{ij}|\theta_{2}\right)}^{\left(1-z_{1i}\times z_{2j}\right)}p_{1}^{z_{1i}}{\left(1-p_{1}\right)}^{1-z_{1i}}\;,\;\mathrm{and}\;z_{2j}^{\left(t\right)}=\int \left\{p_{2}^{z_{2j}}{\left(1-p_{2}\right)}^{1-z_{2j}}\times \overset{J}{\underset{j=1}{\prod }}\int \mathrm{Pr}{\left(y_{ij}|\theta_{1}\right)}^{z_{1i}\times z_{2j}}\mathrm{Pr}{\left(y_{ij}|\theta_{2}\right)}^{\left(1-z_{1i}\times z_{2j}\right)}p_{1}^{z_{1i}}{\left(1-p_{1}\right)}^{1-z_{1i}}dz_{1i}\right\}dz_{2j}$$

We calculate the $E\left(z_{1i},z_{2j}|\theta^{\left(t\right)},Y\right)$ in the **E step** as follow:(6)$$E(z_{1i},z_{2j}|\theta^{\left(t\right)},Y)=\mathrm{Pr}\left(z_{1i}=1,z_{2j}=1|y_{ij};\;\theta^{\left(t\right)}\right)=\frac{Pr\left(z_{1i}=1,z_{2j}=1\right)\mathrm{Pr}\left(y_{ij}|z_{1i}=1,z_{2j}=1;\theta^{\left(t\right)}\right)}{\sum_{i=1}^{I}\sum_{j=1}^{J}Pr\left(z_{1i},z_{2j}\right)\mathrm{Pr}\left(y_{ij}|z_{1i},z_{2j};\theta^{\left(t\right)}\right)}\;=\frac{Pr\left(z_{1i}=1,z_{2j}=1\right)\frac{Pr{\left(y_{ij}|\theta_{1}\right)}^{z_{1i}\times z_{2j}}Pr{\left(y_{ij}|\theta_{2}\right)}^{\left(1-z_{1i}\times z_{2j}\right)}\mathrm{Pr}\left(z_{1i}\right)\mathrm{Pr}\left(z_{2j}\right)}{Pr\left(z_{1i}=1,z_{2j}=1\right)}}{\sum_{i=1}^{I}\sum_{j=1}^{J}Pr\left(z_{1i},z_{2j}\right)\mathrm{Pr}\left(y_{ij}|z_{1i},z_{2j};\theta^{\left(t\right)}\right)}\;=\frac{p_{1}p_{2}{\left(\frac{1}{\sqrt{2\pi }\sigma_{1}\delta_{1}^{}\tau_{1}^{}}exp\frac{-{\left(y_{ij}-\mu_{1}-\alpha_{1i}-\beta_{1j}\right)}^{2}}{2\sigma_{1}^{2}}exp\frac{-\alpha_{1i}{}^{2}}{2\delta_{1}^{2}}exp\frac{-\beta_{1j}{}^{2}}{2\tau_{1}^{2}}\right)}^{z_{1i}\times z_{2j}}}{\sum_{i=1}^{I}\sum_{j=1}^{J}Pr\left(z_{1i},z_{2j}\right)\mathrm{Pr}\left(y_{ij}|z_{1i},z_{2j};\theta^{\left(t\right)}\right)}\;\times \frac{{\left(\frac{1}{\sqrt{2\pi }\sigma_{2}\delta_{2}^{}\tau_{2}^{}}exp\frac{-{\left(y_{ij}-\mu_{2}-\alpha_{2i}-\beta_{2j}\right)}^{2}}{2\sigma_{2}^{2}}exp\frac{-\alpha_{2i}{}^{2}}{2\delta_{2}^{2}}exp\frac{-\beta_{2j}{}^{2}}{2\tau_{2}^{2}}\right)}^{\left(1-z_{1i}\times z_{2j}\right)}}{1}=\gamma \left(z_{1i},z_{2j}\right)$$
where *t* = 1, 2, …, *N* is the number of iterations; *i* and *j* refer to the gene and sample set, respectively, with parameter space of $i\in \left\{1,\ldots ,S_{I}\right\}$ and$\;j\in \left\{1,\ldots ,S_{J}\right\}$.

(iii) The **M step** is used to calculate the maximum log-likelihood $argmax_{\theta ,P}\left(l_{c}\left(Y,Z|\theta^{\left(t\right)},\right)\right)$ with respect to the estimated likelihood function (Equation (7)) probabilities (${\hat{p}}_{1}^{\left(t\right)},{\hat{p}}_{2}^{\left(t\right)}$) of ${\hat{z}}_{1i}\;$and ${\hat{z}}_{2j}$ from the E step. This produces new distributional parameters of the observed data, $\theta^{\left(t+1\right)\;}$, using Equation (8).
(7)$$\begin{array}{ll} l_{Ac}\left(Y,Z;\theta^{\left(t\right)}\right) & =\overset{I}{\underset{i=1}{\sum }}\overset{J}{\underset{j=1}{\sum }}\left(z_{1i}\times z_{2j}\right)log\left(Pr\left(y_{ij}|\theta_{1}\right)\right)+\left(1-z_{1i}\times z_{2j}\right)\log\left(Pr\left(y_{ij}|\theta_{2}\right)\right)+\log\left(Pr\left(z_{1i}\right)\right)\\ & +\log\left(Pr\left(z_{2j}\right)\right)\\ & =\overset{I}{\underset{i=1}{\sum }}\overset{J}{\underset{j=1}{\sum }}\left(z_{1i}\times z_{2j}\right)\times (\log(\frac{1}{\sqrt{2\pi }\sigma_{1}{}^{\left(t\right)}\delta_{1}{}^{\left(t\right)}\tau_{1}{}^{\left(t\right)}})-\frac{{\left(y_{ij}-\mu_{1}{}^{\left(t\right)}-\alpha_{1i}{}^{\left(t\right)}-\beta_{1j}{}^{\left(t\right)}\right)}^{2}}{2\sigma_{1}^{2\left(t\right)}}-\frac{\alpha_{1i}{}^{2\left(t\right)}}{2\delta_{1}^{2\left(t\right)}}\\ & -\frac{\beta_{1j}{}^{2\left(t\right)}}{2\tau_{1}^{2\left(t\right)}})\\ & +\left(1-z_{1i}\times z_{2j}\right)\times (\log\left(\frac{1}{\sqrt{2\pi }\sigma_{2}{}^{\left(t\right)}\delta_{2}{}^{\left(t\right)}\tau_{2}{}^{\left(t\right)}}\right)-\frac{{\left(y_{ij}-\mu_{2}{}^{\left(t\right)}-\alpha_{2i}{}^{\left(t\right)}-\beta_{2j}{}^{\left(t\right)}\right)}^{2}}{2\sigma_{2}^{2\left(t\right)}}-\frac{\alpha_{2i}{}^{2\left(t\right)}}{2\delta_{2}^{2\left(t\right)}}-\frac{\beta_{2j}{}^{2\left(t\right)}}{2\tau_{2}^{2\left(t\right)}})\\ & +z_{1i}log{\hat{p}}_{1}^{\left(t\right)}+\left(1-z_{1i}\right)log\left(1-{\hat{p}}_{1}^{\left(t\right)}\right)+z_{2j}log{\hat{p}}_{2}^{\left(t\right)}+\left(1-z_{2j}\right)log\left(1-{\hat{p}}_{2}^{\left(t\right)}\right) \end{array}$$

Based on the E step, the posterior distribution $\gamma \left(z_{1i},z_{2j}\right)$ is calculated. By setting derivatives of the log-likelihood function $l_{Ac}\left(Y,Z;\theta^{\left(t+1\right)}\right)$ (7) to zero with respect to parameters$\;\mu_{k}^{\left(t\right)},\alpha_{k}^{\left(t\right)},\beta_{k}^{\left(t\right)},\;and\;\sigma_{k}^{\left(t\right)}$, the final estimates of parameters $\theta_{1}^{\left(t+1\right)}\;$and group membership probabilities (${\hat{p}}_{1}^{\left(t+1\right)},{\hat{p}}_{2}^{\left(t+1\right)}$) are updated as follows:$$\beta_{1j}^{\left(t\right)}=\frac{\sum_{ij}\gamma^{\left(t\right)}\left({\hat{z}}_{1i},{\hat{z}}_{2j}\right)\tau_{1}^{2\left(t\right)}\left(y_{ij}-\mu_{1}^{\left(t\right)}-\alpha_{1i}^{\left(t\right)}\right)}{\sum_{ij}\gamma^{\left(t\right)}\left({\hat{z}}_{1i},{\hat{z}}_{2j}\right)\tau_{1}^{2\left(t\right)}\sigma_{1}^{2\left(t\right)}},$$
$$\alpha_{1i}^{\left(t\right)}=\frac{\sum_{ij}\gamma^{\left(t\right)}\left({\hat{z}}_{1i},{\hat{z}}_{2j}\right)\delta_{1}^{2\left(t\right)}\left(y_{ij}-\mu_{1}^{\left(t\right)}-{\hat{\beta }}_{1j}^{\left(t\right)}\right)}{\sum_{ij}\gamma^{\left(t\right)}\left({\hat{z}}_{1i},{\hat{z}}_{2j}\right)\sigma_{1}^{2\left(t\right)}\delta_{1}^{2\left(t\right)}},$$
(8)$${\hat{\mu }}_{1}^{\left(t+1\right)}=\frac{\sum_{ij}\gamma^{\left(t\right)}\left({\hat{z}}_{1i},{\hat{z}}_{2j}\right)\left(y_{ij}-{\hat{\alpha }}_{1i}^{\left(t+1\right)}-{\hat{\beta }}_{1j}^{\left(t+1\right)}\right)}{\sum_{ij}\gamma^{\left(t\right)}\left({\hat{z}}_{1i},{\hat{z}}_{2j}\right)},\;{\hat{\sigma }}_{1}^{2\left(t+1\right)}=\frac{\sum_{ij}\gamma^{\left(t\right)}\left({\hat{z}}_{1i},{\hat{z}}_{2j}\right){\left(y_{ij}-{\hat{\mu }}_{1}^{\left(t+1\right)}-{\hat{\alpha }}_{1i}^{\left(t+1\right)}-{\hat{\beta }}_{1j}^{\left(t+1\right)}\right)}^{2}}{\sum_{ij}\gamma^{\left(t\right)}\left({\hat{z}}_{1i},{\hat{z}}_{2j}\right)},\;\delta_{1}^{2\left(t+1\right)}=\frac{\sum_{ij}\gamma^{\left(t\right)}\left({\hat{z}}_{1i},{\hat{z}}_{2j}\right){\hat{\alpha }}_{1i}^{\left(t+1\right)}}{\sum_{ij}\gamma^{\left(t\right)}\left({\hat{z}}_{1i},{\hat{z}}_{2j}\right)},\;\tau_{1}^{2\left(t+1\right)}=\frac{\sum_{ij}\gamma^{\left(t\right)}\left({\hat{z}}_{1i},{\hat{z}}_{2j}\right){\hat{\beta }}_{1i}^{\left(t+1\right)}}{\sum_{ij}\gamma^{\left(t\right)}\left({\hat{z}}_{1i},{\hat{z}}_{2j}\right)},\;{\hat{p}}_{1}^{\left(t+1\right)}=\frac{\sum_{i}{\hat{z}}_{1i}^{\left(t\right)}}{I},{\hat{p}}_{2}^{\left(t+1\right)}=\frac{\sum_{j}{\hat{z}}_{2j}^{\left(t\right)}}{J}\;,\;\;$$
where *t*, *i*, and *j* refer to the number of iterations, the gene set, and the sample set, respectively, as in (8). $\theta_{2}^{\left(t+1\right)}$ can be similarly estimated.

(iv) **Convergence criteria**: The algorithm stops when the relative change in the log-likelihood is sufficiently small.
(9)$$l_{Ac}\left(Y,Z;\theta^{\left(t+1\right)}\right)-l_{Ac}\left(Y,Z;\theta^{\left(t\right)}\right)<\epsilon$$
where ϵ is a suitably small value specified by the user, and the default value is 0.00001.

### 2.3. Extracting Members of the Bicluster

The EM algorithm fits the empirical Bayes biclustering model and produces a probability matrix of the biclustering membership $p_{ij}=E\left(z_{1i}z_{2j}|y_{ij}\right)$, which is associated with each data point $y_{ij}\;$in matrix Y. Each $p_{ij}$ in matrix P indicates the probability of data point $y_{ij}$ belonging to the bicluster. The algorithm constructs the potential bicluster from matrix Y under probability *P* according to the following conditions: (i) the bicluster contains as many data points as possible under a certain probability threshold; (ii) the bicluster includes points with different conditions, such as tumor samples and cancer cell lines; (iii) special genes have a higher probability to members of the bicluster. Typically, drug target genes, oncogenes, or tumor suppressers are of interest. The algorithm requires user-defined three parameters in order to search for the specific bicluster. (i) The probability threshold$\;Ac\in \left[0,1\right]\;$denotes the probability of data point $y_{ij}$ being selected in the bicluster or not based on a *sign* function$\;p_{c}=\{1,\;if\;p\left(z_{1i}=1,z_{2j}=1|y_{ij}\right)>Ac;\;0,\;if\;p\left(z_{1i}=1,z_{2j}=1|y_{ij}\right)<Ac\}$. $p_{c}$ = 1 indicates a ‘yes’ membership of bicluster, and 0 otherwise. Parameter *A**c* is a sensitivity parameter. A high-value *c* increases the accuracy of the bicluster, but results in fewer genes and samples in a bicluster. (ii) The average of probability values in the bicluster, $p_{ave}\in \left[0,1\right]$, is used to decide the bicluster block size and overall accuracy of the bicluster. The higher $p_{ave}\;$is, the higher accuracy of the bicluster. For example, if we select *A**c* = 0.8 and $p_{ave}$ = 0.95, the bicluster has a 80% probability threshold of membership, and its overall bicluster accuracy is 95%. (iii) A pre-speculated seed is needed as a starting point to search a bicluster. The default seed is the center of matrix.

### 2.4. Bicluster-Searching Algorithm after the Bi-EB Algorithm

Figure 1d illustrates the bicluster-searching algorithm when Bi-EB is under the parameter setting Ac,$\;p_{ave}$, and seeds. After the EM algorithm converges, the probability matrix *P* = {$p_{ij}$} of the bicluster membership is calculated and assigned to each data point. *P* is then transformed into a binary matrix *U* based on the probability threshold$\;p_{c}$, in which $\;u_{ij}$ is 1 if gene *i* and condition *j* belong to the bicluster, and 0 otherwise. Genes and samples with the highest number of 1s will be arranged to the right corner of matrix U. If samples are from two different groups (i.e., tumor samples and cell lines), the sorting process will be applied separately on them. The final bicluster *B* is defined by a sub-matrix of *U**,*** in which at least $p_{ave}$ of the elements is equal to 1. The default value of $p_{ave}$ is 0.95 here. In other words, 95% of data points in the constructed bicluster have a bicluster membership probability higher than the threshold. Table 1 lists the Bi-EB algorithm process.

### 2.5. Performance Comparisons among Biclustering Algorithms

To evaluate the Bi-EB algorithm’s performance, we compare seven different bicluster algorithms to ours. The seven algorithms, including Cheng and Church (CC) [2], Plaid [4], xMOTIFs [22], BiMax [23], Spectral [24], FABIA [25], and QUBIC [26], are used by Eren [27] and Peng Sun [28]. To keep the data analyses and results consistent, all algorithms are carried out in a ‘biclust’ R package. More information on algorithms and related papers can be found in Table 2. Each simulation run 10 times, and the evaluation of performance is based on their average result. 

The performance of bicluster algorithms heavily depends on their parameter setting. In order to optimize the biclustering result, parameters are specifically set for each synthetic data in multiple steps. In each algorithm, its parameters are given a vector of values. A final value is chosen based on the performance of the algorithm under that value and its combination with other parameters. The measurements used to evaluate the performance of each simulation are discussed in the following sections.

#### 2.5.1. Synthetic Data Generation

Three synthetic datasets with known patterns of biclusters are used to evaluate the Bi-EB model’s performance. They include constant, row scale-shift, and column scale-shift biclusters. The performances of Bi-EB and seven other algorithms are compared for these simulation data. All simulations are run in R version 3.2.4 and RStudio version 0.99.902.

Three synthetic datasets of size 200 × 300 with one embedded bicluster are simulated. (i) The constant bicluster has a size of 25 × 25 with the standard Gaussian distribution (i.i.d.) *N*(10,1). The background *noise* is randomly chosen from the Gaussian distribution of *N*(4,1) independently. (ii) The row scale-shift bicluster has a size of 70 × 70 with scaled-shifted rows. The bicluster rows are shifted and scaled from base row$\;R_{i}\sim N\left(0,1\right)$. Each row is shifted and scaled using formula$\;N\left(5,1\right)+N\left(5,1\right)*R_{i}$. The background noise is drawn from the standard Gaussian distribution (i.i.d.) N(0,1). (iii) The column scale-shift bicluster has a size of 70 × 70 with a scaled-shifted column. The bicluster pattern and formula are similar to the row shift-scale bicluster, and they are only applied to columns instead of rows. Synthetic datasets (see Appendix A) are used as an input matrix of eight biclustering algorithms, including our Bi-EB model.

#### 2.5.2. Evaluation Measurements

The algorithms’ performances on synthetic data are evaluated by comparing extracted biclusters with pre-defined biclusters. We follow methods proposed by Eren et al. [27] and Peng Sun [28] to score the biclusters. Three measurements of *Jaccard coefficient*, *recovery*, and *relevance* are used for this comparison. The Jaccard index indicates the relative overlap between two biclusters. Let *b*_1_ and *b*_2_ be two biclusters, and the score *s* is defined to compare two biclusters by the function:(10)$$s\left(b_{1},b_{2}\right)=\frac{\left|b_{1}⋂b_{2}\right|}{\left|b_{1}\cup b_{2}\right|}$$
where $\left|b_{1}⋂b_{2}\right|\;$is the number of data elements in their intersection and $\left|b_{1}\cup b_{2}\right|$ is the number of elements in their union. The maximum value of 1 indicates two identical biclusters, and the minimum value of 0 represents two non-overlapping biclusters.

The recovery function refers the comparison between the true bicluster *T* in the simulation model and bicluster *R*, estimated from the data. The recovery function is defined by:(11)$$S\left(T,R\right)=\frac{1}{\left|T\right|}\underset{b_{1}\subseteq T}{\sum }\underset{b_{2}\in R}{\max}s\left(b_{1},b_{2}\right)$$

The score ranges from 0 to 1. A recovery score is interpreted as the percentages of the true set that is recovered from the bicluster analysis results. 

Relevance is another function used to evaluate the similarity between a true bicluster and a bicluster result set. The relevance function is calculated by:(12)$$S\left(T,R\right)=\frac{1}{\left|R\right|}\underset{b_{1}\subseteq R}{\sum }\underset{b_{2}\in T}{\max}s\left(b_{1},b_{2}\right)$$

Similar to recovery, relevance score ranges between 0 and 1. If all the bicluster result sets are in the true set, meaning *R* ⊆ *T*, the score is 1. The relevance score indicates the percentage of the bicluster result set that is shared with the true biclusters.

#### 2.5.3. The Bi-EB Algorithm on the Three Synthetic Datasets

The Bi-EB parameter ‘*Ac*’ is set as 0.8 and 0.6 for constant-, row*-*, or column (*col*)-scales biclusters, respectively. Parameter ‘$p_{ave}$’ is set as 0.95 for the constant and 0.7 for row/col-scaled data. Figure 2 shows the simulation results using Bi-EB on three synthetic datasets. Figure 2(a1–a4) focuses on the constant shift pattern. A histogram of synthetic data with a constant pattern shows the fitted Gaussian mixture model on the background noise and the Bi-EB bicluster (*curve*) in Figure 2(a1). The position of the bicluster is observable from the data heatmap in Figure 2(a2). The initial values of *θ* are set based on the estimates from one time run of the likelihood function. With the initial values of $\;p_{1}^{\left(0\right)},p_{2}^{\left(0\right)}=$ 0.5, the parameters $\;\mu_{1},\mu_{2},\sigma_{1},\sigma_{2}$ are estimated by running the likelihood function. Then, *θ* is given a vector of 4 values for each parameter$\;\mu_{1},\mu_{2},\sigma_{1},\sigma_{2}$ around the estimated values. The vector is (−5, 5) for $\;\mu_{1}\;$and $\mu_{2}\;$and is (−2, 2) for $\sigma_{1}$ and $\sigma_{2}$. The algorithm shows robustness to initial values of$\;\mu_{1},\mu_{2},\sigma_{1},\sigma_{2}$ because the EM algorithm converges to the true values in all the cases. In the constant pattern simulation datasets, our algorithm detects 100% of true bicluster points. In the row-scaled pattern simulation datasets, by setting the parameter ‘*A**c*’ to 0.6, a recovery score of 1 is achieved. Figure 2(b1–b4) illustrates the Bi-EB algorithm data process for the row-scaled data. A histogram of the mixture row-scaled data shows a long tail for the normal distribution which contains bicluster points. An initial value of *θ* is kept the same as the constant pattern. The Bi-EB algorithm successfully reports a recovery score of 0.9 and a relevance score of 0.989. A similar simulation in col-scaled data results in a recovery score of 1. As Figure 2(c1–c4) indicates, the algorithm converges to true values with the same initial values as the constant- and row-scaled data. 

To further investigate the impact of parameter settings in bicluster identification, we compare simulation results over a vector of values for *A**c* and$\;p_{ave}$ (Figure 2d,e) under the row-scaled synthetic data. Different values of *A**c* are chosen from (0.2 to 0.8) in implementing our Bi-EB algorithm. When *Ac* is 0.2, the recovery score identified by the Bi-EB algorithm is 0.85. Figure 2(e4) shows the ratio of the extracted bicluster members over embedded true bicluster points. When this ratio meets one, the resulted bicluster has no false positive or negative noise. The final bicluster with $p_{ave}$ of 0.5 includes 45% noise points which are not true bicluster members. The number of false-positive points in the extracted bicluster decreases as the value of $p_{ave}$ increases. In our simulation, with $p_{ave}$ = 0.7, the ratio of extracted bicluster members over true bicluster points is one. However, increasing $p_{ave}$ from 0.75 to 0.95 using the Bi-EB algorithm obtains smaller biclusters.

#### 2.5.4. Comparison Based on Evaluation Measurements

In order to evaluate the performance of our algorithm, we compare our biclustering results with seven other bicluster algorithms. We adopt the comparison framework recommended by Eren [27] and Sun [28]. The bicluster results are compared based on their recovery and relevance scores on 3 synthetic datasets. Each bicluster algorithm is ran on 10 different simulation datasets under each of three simulation settings, and the averages of recovery and relevance scores are compared. Figure 3 compares the recovery and relevance of the seven algorithms and demonstrates their bicluster heatmaps under three simulation patterns. In the constant pattern simulation setting, the Bi-EB algorithm has a recovery score and a relevance score of 1. Please note that even though CC is designed to find constant bicluster patterns, it fails to find the true constant bicluster in the simulation study, i.e., the recovery and relevance score are approximately 0.5. In finding the constant bicluster, both the xMotif and spectral method gave a recovery and relevance score of zero. On the other hand, BiMax, FABIA, Plaid, and QUBIC perform well in finding a constant bicluster with a recovery and relevance score of 1. 

On the other hand, as for the parameter setting to Bi-EB, as Figure 2 (d1–d4) indicates, increasing *Ac* results in higher percentages of identified true points. The highest recovery value of 0.99 is observed when a *Ac* value reaches 0.8 and stays constant afterwards. However, at *c =* 0.8, a recovery score drops by 5% compared to *c* = 0.6. Thus, we chose *Ac*
*=* 0.6 which extracts higher percentages of true points with a recovery score of 0.98. Next, we fix the value of *Ac* at 0.6 and change $p_{ave}$ from 0.5 to 0.95. The Bi-EB algorithm identifies a large number of background data as bicluster points with smaller values of$\;p_{ave}$. 

In the row/column scale-shifted simulation datasets, our algorithm outperforms all other algorithms in finding scale-shifted row and column biclusters. Bi-EB has a recovery and relevance score of 1. In row scale pattern data, BiMax, Plaid, and xMotif algorithms have recovery scores of 0.008, 0.0183, and 0.002, respectively. In the column scale-shifted simulation, the Cheng and Church (CC) algorithm has a recovery score of 0.6, while the resulted biclusters in row scale-shifted data using the CC algorithm are very poor (recovery score = 0.0011 and relevance score = 0.00014). The FABIA can find 18% of true biclusters (recovery score = 0.18) with a scale-shifted pattern in rows, whereas the result with the same pattern in columns only has a recovery score of 0.07. QUBIC also fails to find scale and shift patterns in either row or column (with the average of recovery and relevance around 7%). Spectral only has a relevance score of 2% and a recovery score of 30%.

## 3. Results

We develop a novel biclustering algorithm using the empirical Bayes mixture model, called Bi-EB. The model can search for a bicluster in two directions simultaneously: multi-omics data (i.e., the mRNA gene expression (GE) and the protein amount (PA)) and multi-conditional samples (such as tumors and cancer cell lines). The bicluster is characterized through a hierarchical model structure built upon normal distributed log-transformed ratios between mRNA expressions and protein expressions (GEs/PAs). Bi-EB is used to search these block clusters from the ratio matrix across multi-conditional samples.

### Bi-EB Targets the Module Detection of Common mRNA Expression/Protein Amount on Breast Cancer

The Bi-EB algorithm is used to seek shared molecular profiles in order to facilitate the translational research between breast cancer cell lines and tissue samples in different subtypes on GE/PA ratio from both cancer cell lines and tissue samples. Here, it is our best interest to use the Bi-EB algorithm to identify shared GE/PA ratios to seek common variation patterns in luminal A/B and basal-like subtype breast cancer cell lines and tumors. Gene expression data are derived from TCGA RNA sequencing, and the protein amount is taken from the reverse-phase protein array (RPPA).

The TCGA breast cancer dataset is obtained from the Broad Institute GDAC Firehose (https://gdac.broadinstitute.org/, accessed on 1 March 2022). The protein amount is taken from the RPPA data, an antibody-based protein assay platform, which is compared to the number of gene features in the gene-level data.

The Bi-EB searching algorithm allows bicluters that share a gene (i.e., drug target genes) to be found. The iterative Bi-EB bicluster selection requires initial values for two parameters: the bicluster membership probability threshold Ac and the bicluster average probability p. They were set to 0.8 and 0.9, respectively. In our sensitivity analysis, changing Ac to 0.7 or 0.9 did not change resulted biclusters much. We set this seed to be the default value, which is the center of the matrix.

(i)The luminal A/B subtype

Luminal A/B is one the well-known subtypes of breast cancer with the most successful targeted therapy drugs comparing to the other subtypes. Since the ER status of luminal subtype is positive, we expect to see ER-α’s GE/PA ratio in our bicluster. Therefore, the bicluster containing ER-α’s GE/PA in the luminal subtype shall serve as a positive control in our Bi-EB analysis. The input data of the Bi-EB algorithm represent a matrix of 45 gene GE/PA ratio with 279 samples (CCLE breast cancer 10 cell lines and the *Cancer Genome Atlas* breast invasive carcinoma (TCGA-BRCA) data collection 269 tissue samples [10,11]). 

Since cell line and tissue measurements have different platforms, we normalize both protein and mRNA expressions separately for cell line and tumor. To explain the Bi-EB algorithm, hierarchical clustering is used to cluster each mRNA and RPPA protein cluster in tumors and cancer cells separately, as shown in Figure 4(a1,a2,b1,b2). The order of genes and samples are kept for mRNA expression heatmap (Figure 4). The RPPA expression pattern in the tumor does not show any relation to its corresponding mRNA pattern in the cancer cell, while by using the ratio co-variation of the mRNA/RPPA Bi-EB algorithm, the hidden patterns between tumor and cancer cells are identified. The Bi-EB algorithm identifies the common ratio pattern of mRNA/RPPA from cancer cells and tumors. Figure 4(a3,b3) show the bicluster heatmap, clearly indicating the shared GE/PA ratios between cell lines and tumor samples. The black frame represents the bicluster shared GE/PA ratios between cancer cells and tumors. The largest bicluster estimated by Bi-EB includes 10 cell lines and 242 tissue samples in a subgroup of 12 genes. The identified cell lines in bicluster are BT474, BT483, CAMA1, HCC1428, HCC2218, MCF7, MDAMB361, MDAMB415, MDAMB453, and T47D. The genes include ESR1, ERBB2, AR, GATA3, CDH1, KDR, BCL2, RAB25, PDK1, COL6A6, PRKCA, and CASP7.

We further investigate the gene expression co-variations between GE and PA within cancer cells and tumors extracted by our Bi-EB algorithm (Figure 5). Figure 5a illustrates the GE/PA ratios between cell lines and tumor tissues. The ratios appear to be constant within either cell lines or tumor tissue samples. Cell lines have a higher ratio than the tumor tissue samples. Figure 5a further shows that there are two genes (PDK1 and PRKCA) whose GE/PA ratios are lower than other genes in the bicluster. We use ESR1 as an example. Figure 5b shows the ESR1 GE/PA ratio across 10 cell lines and 242 tissue samples. The ratio ranges from −0.5 to 2.3. In Figure 5c, both protein and mRNA levels of ESR1 are plotted, and they appear to be correlated. This correlation pattern is further clarified in Figure 5d. In Figure 5a, a closer look at 100 samples further indicates that the mRNA level of ESR1 is a merely shifted pattern of the ER protein level. The other identified genes in the bicluster such as AR, BCL2, ERBB2, and CDH1 have shown similar significant correlation between the mRNA and the protein in the cell line and tissue [29]. Based on the DrugBank database, AR, BCL2, ERBB2, and ESR1 are known drug targets for breast cancer patients. Out of 12 GE/PA ratios in the bicluster, 9 mRNA expressions are significantly correlated with their protein expression (r > 0.5, *p* < 0.05) as potential targets.

(ii)The Basal-like subtype

In the basal-like subtype breast cancers, they are usually triple-negative, i.e., they lack the estrogen receptor (ER), the progesterone receptor (PR), and the human epidermal growth factor receptor 2 (HER2) [16]. Due to the heterogeneity of the disease, this breast cancer subtype has short survival and shows a poor response to either hormone therapies or HER2-targeted therapies [30]. The absence of well-defined molecular targets has been the primary challenge in treating TNBC breast cancer patients. In the basal-like group, 68 tissue samples and 15 cell line samples and their 45 GE/PA were analyzed for bicluster analysis. The biclustering result is displayed in Figure 4(b3). The largest bicluster contains 10 genes across 11 cell lines and 62 tissue samples. The genes in this bicluster include CCNB1, CDH1, KDR, RAB25, COL6A1, COL6A2, COL6A6, ERBB2, PGR, and PRKCA. Figure 4(b1,b2) illustrates the membership probabilities of the extracted bicluster. Identified cell lines by Bi-EB are expected to be good representatives for basal-like tumor tissues. These cell lines include BT549, HCC1143, HCC1395, HCC1569, HCC1599, HCC1806, HCC38, HCC70, MDAMB157, MDAMB231, and MDAMB468. As this bicluster includes parameter Ac = 0.8 and p = 0.9, at least 90% of its data elements have bicluster membership probabilities larger than 0.8. 

Then, we further explore the variations in the GE/PA ratios of this bicluster. The overall GE/PA ratio pattern of these 10 genes from 83 cell lines and tissue samples is shown in Figure 6a. Similar to the luminal breast cancer subtype, in the basal-like subtype, cell lines have higher GE/PA ratios than tumor tissues. Interestingly, genes PR and PRKCA have lower ratios than the other genes. We decide to focus on two genes, CCNB1 and RAB25. RAB25 is located at chromosome 1q22. It is amplified at the DNA level and overexpressed at the RNA level in the breast cancer. These changes correlate with a worsened outcome in both diseases. In addition, overexpressed RAB25 in breast cancer cells decreases the apoptosis and increases the proliferation and aggressiveness in vivo. The CCNB1 protein level has been shown to differ among breast cancer subgroups [20] and different histological grades [20,21]. It was also associated with breast cancer outcomes [20,21,31]. In addition, CCNB1 is included in several prognostic gene signatures, such as the 21-gene recurrence score [12] and two other genomic signatures [32,33].

Therefore, our Bi-EB algorithm uses genes CCNB1 and RAB25 as examples. A covariation of the ratio for CCNB1 (golden line) in Figure 6a shows a noticeable difference in the mRNA expression as compared to protein in cell lines, while this difference is smaller in tissue. Figure 6b,c show this difference in detail. The ratio level of CCNB1 keeps low values for tissue samples and causes a sudden increase in samples of cancer cell lines, where mRNA measurements rise faster than protein expressions. It is evident from Figure 6c that the protein level of CCNB1 is higher than mRNA across the majority of samples, even though this scenario changes the pattern of variation in cancer cell lines and tumors for a subset of conditions. A closer look at the pattern in Figure 6d justifies the constant ratio of this gene. Figure 6e–g illustrate the relations of mRNA and the proteins of RAB25. Similar to CCNB1, the ratio varies in a small range for the majority of samples and increases from 1 to 3.5 in cells for 10 samples. The expression measurements of mRNA with a slight difference are constantly higher than protein RAB25, resulting in a positive ratio of mRNA–protein in tumors, except when the mRNA rises to 9 and the protein amount stays between 0 and 2 in the cancer cell lines.

The genes selected by the Bi-EB algorithm are important genes in the basal-like subtype. In this intrinsic subgroup, CCNB1, CDH1, KDR, RAB25, and PRKCA have shown significant correlations of mRNA-RPPA with folder change>1.5, r>0.5, and p<0.05 [29]. According to the DrugBank database, RAB25 and ERBB2 are known drug targets.

## 4. Discussion

Input data preparation is a crucial step in biclustering algorithms. It is important to use appropriate normalizing method that suits the integration of multiple-omics data analysis [20,34]. Bi-EB using a unique scaling ratio to detect the variations in multiple-omics data, which has not been discussed in many biclustering research studies before. Bi-EB tends to find biclusters with tightly co-expressed mRNA expressions and proteins across cancer cells and tumors, in which the biclusters are markedly a ratio of GE/PA, both of cancer cells and tumors.

Bi-EB is an automatic feature detection model on a pixel level (dot level) in a matrix, where the dot variation in the matrix is a joint variation from its associated row and column variation (marginal variation). Two important parameters (*p_ave_*: the accuracy of a bicluster and *Ac*: the probability of belonging to a bicluster) are designed to regulate the joint and marginal variations and obtain the optimal bicluster. At first, the Bi-EB method uses an EM algorithm, and conducts a data-driven search for bicluster patterns in the expression data. The algorithm outputs the probability of each data point belonging to a potential bicluster. If we set *Ac* = 0.85 as the probability to assign a gene to the bicluster, all genes in the bicluster shall have an 85% membership probability or higher. Bi-EB is used for a dot possibility calculation in a bicluster. Hence, its speed is not as fast as other biclustering models, which will be improved in future. 

In applying our Bi-EB model to seek biclusters from breast cancer transcriptome data and protein data, we address two innovative cancer biology questions. Firstly, to our knowledge, for the first time, we specifically investigate possible shared biclusters between cell lines and primary breast cancer tumors to answer the question on whether breast cancer lines are reasonable models for breast tumors. Breast cancer cell lines that are in one bicluster with large tumor samples shall be good representatives of tumor tissue. Secondly, in our bicluster analysis, the mRNA expression and protein abundance ratio, i.e., GE/PA, is modeled, instead of the mRNA expression or protein abundance themselves. This GE/PA allows us to understand whether the transcription signal will be translated into the protein level. In the era of cancer precision medicine, many current drug target selection strategies are based on genomic variants and transcriptome data, as opposed to proteomic data. However, most drugs are targeted on proteins, as opposed to gene mRNA. Therefore, a better understanding of the consistency between mRNA expression and protein abundance may improve our confidence in drug target selection.

The extracted biclusters in basal-like and luminal subtypes appear across the cell line and tumor samples. The cell lines found in the bicluster are expected to exhibit the heterogeneity of primary breast tumors. In our study, in luminal A, the cell lines that are found to be a good model of tumor samples are BT474, BT483, CAMA1, HCC1428, HCC2218, and MCF7. In the basal-like subtype, based on the bicluster, the identified cell lines are BT549, HCC1143, HCC1395, HCC1569, HCC1599, HCC1806, HCC38, and HCC70. In the breast cancer cell line study by Jiang et al. [13], the characteristics of 51 cell lines in relation to the primary breast tumors were investigated. Based on their results, BT474, CAMA1, HCC1428, BT483, and MCF7 showed to mirror the biological features of tumor samples to a great extent. The same study shows that the BT549, HCC1143, HCC1569, and HCC70 cell lines represent the genomic features of breast tumor in the basal-like subtype. 

We utilize the Bi-EB algorithm to determine genes with a shared pattern across the subgroup of TCGA and CCLE breast cancer samples. From our results, extracted genes in the basal-like bicluster are all highly correlated (*p*-value < 0.05) with their related protein. In a recent study, Li et al. [31] performed UQ-pgQ2 and *DESeq2* and identified differential mRNA expressions in TNBC patients. Some of the genes reported in Li’s study, such as KDR, COL6A6, PGR, and CCNB1, were also found in our basal-like bicluster. KDR, COL6A6, and PGR are identified as differentially down-regulated expressed genes in TNBC, while CCNB1 is identified as a significant up-regulated gene [21]. A heatmap of the basal-like bicluster in our study corresponds to these findings.

Currently, data integration approaches used to efficiently identify subtypes’ genomic variations among existing samples have recently gained attention. Tensor factorization and multi-view correlation analysis methods were applied for either dimension reduction or clustering to provide more amenable data representations for cancer classification and fusion patterns across multi-omics data types among different cancer types [31,32], such as moCluster [33], iCluster [35], and iClusterPlus [36]. These methods, while they are all powerful in detecting shared patterns across multi-omics data, they are not designed to find a shared pattern between cancer cell lines versus patient tumor samples, or to seek translational signals from the transcriptome to the proteome. Our proposed Bi-EB method uniquely positions us to seek gene biclusters that are shared between cell lines and tumor samples, and notably translated signals from mRNA to proteins. As some of the genes identified in two biclusters are also druggable targets among breast cancer subtypes, Bi-EB can be very effective in precision medicine target and drug selection research.

## 5. Conclusions

The genome molecular features shared between cell lines and tumors offer valuable insight into discovering potential drug targets for cancer patients. Our previous studies demonstrate that these important drug targets in breast cancer, ESR1, PGR, HER2, EGFR, and AR have a high similarity in mRNA and protein variations in both tumors and cell lines [13,29]. Based on previous studies, we made a specific hypothesis that there are translational gene sets that are characterized by highly correlated molecular profiles among RNA and proteins. There are translational gene sets that are shared between tumor tissues and cancer cell lines. These gene sets show similar pattern in a subgroup of cell lines and tissue samples. In this study, we aim to integrate cell lines and tissue RNA and protein profiles to characterize drug-able target expression alterations across both RNA and protein data by using the biclustering method. Here, our Bi-EB method based on empirical Bayesian can detect the local pattern of integrated omics data in cancer cell lines versus patient tumor samples. We adopt a data-driven statistics strategy by using the expectation–maximization (EM) algorithm to extract the foreground bicluster pattern from its background noise data in an iterative search. Our novel Bi-EB statistical model has a better chance of detecting co-current patterns of gene and protein expression variations than the existing biclustering algorithms and can seek the drug targets’ co-regulated modules.

## Figures and Tables

**Figure 1 genes-13-01982-f001:**
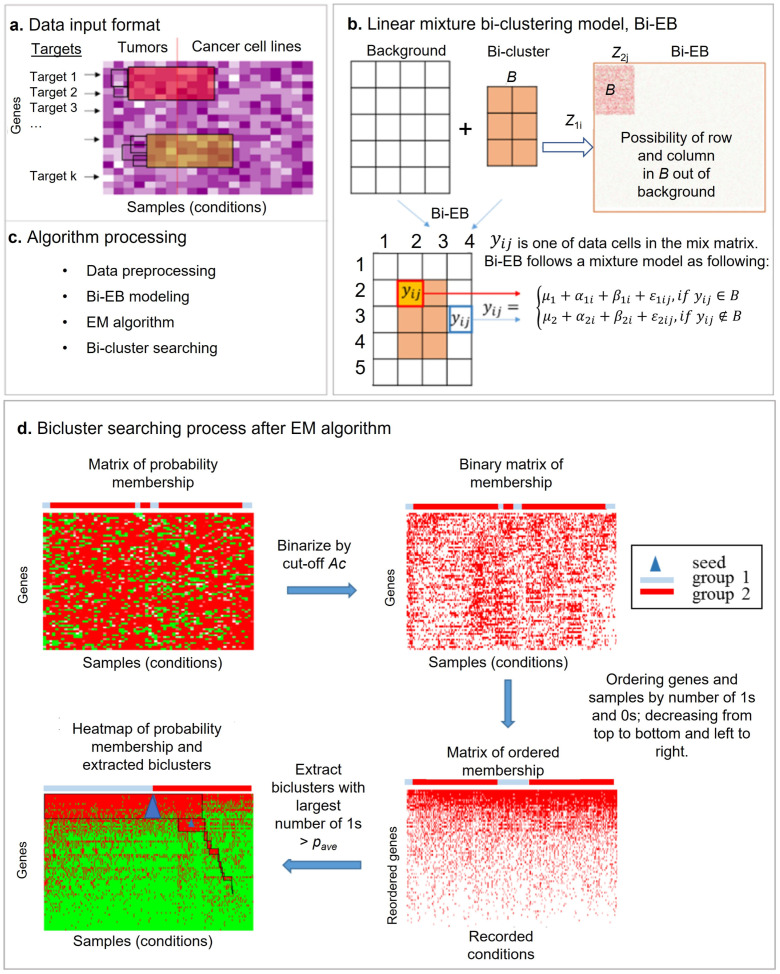
The empirical Bayes model is used to identify the co-regulation biclusters across tumors and cancer cells, both for target module detection. (**a**) Input data for the Bi-EM algorithm (the row is the gene list, and the column is sample list from different groups or conditions); (**b**) linear mixture biclustering model illustration (Bi-EM), where the bicluster signals are extracted from background-originating rows and columns, respectively. The mixture model is constructed to identify these biclusters where its grand mean *μ_1_* is the difference from the background mean *μ_2_* significantly. (**c**) Four processing steps of the Bi-EB algorithm. (**d**) The Bi-EB algorithm searching process. We need to calculate the row and the column possibility in a bicluster and denote each pixel (dot in matrix) in the bicluster as 1 (yes, ≥Ac or 0 (not, <Ac) by cut-off Ac. The Bi-EB algorithm can identify multiple biclusters sequentially with the associated seed. Each iteration can only identify one bicluster. The bicluster size is based on the number of 1s $>p_{ave}$ (the average possibility of row and column in biclusters). The next bicluster search is based on the left information of rows and columns (the current bicluster outside).

**Figure 2 genes-13-01982-f002:**
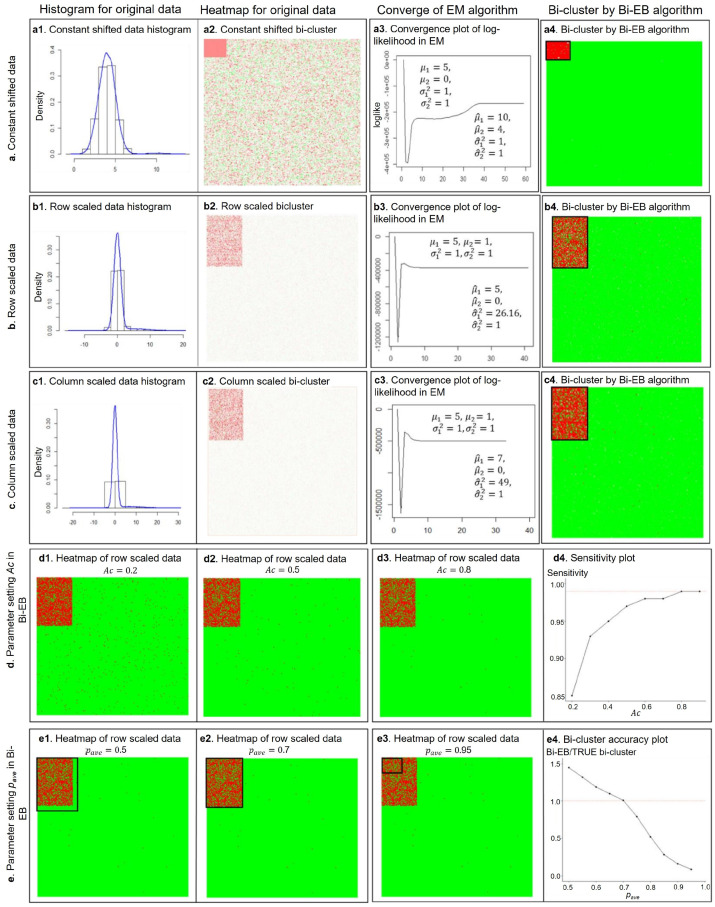
A Bi-EB algorithm for the bicluster on three simulation datasets. (**a**) The Bi-EB algorithm is tested for the *constant-shifted* bicluster pattern. (**a1**,**a2**) displays a histogram plot and heatmap of the original constant shift bicluster data; (**a3**) plots the log-likelihood convergence in the EM procedure of Bi-EB in iteration; (**a4**) displays the extracted bicluster from background using the Bi-EB algorithm. (**b**) The Bi-EB algorithm is tested on a *row-scaled* bicluster pattern. (**c**) The Bi-EB algorithm is tested on *column*-*scaled* bicluster data. (**b1**–**c4**) have the same description as in (**a**). (**d**) The parameter setting of *Ac* in Bi-EB. (**d1**–**d3**) are heatmaps of Bi-EB results with three different values of *Ac*. (**d4**) is the sensitivity plot of the Bi-EB algorithm, while the value of *Ac* changes from 0.2 to 0.09. (**e**) The parameter setting of *p_ave_*. (**e1**–**e3**) are heatmaps of extracted bicluster and (**e4**) is the accuracy plot of Bi-EB biclusters when the *p_ave_* parameter is set from 0.5 to 0.95.

**Figure 3 genes-13-01982-f003:**
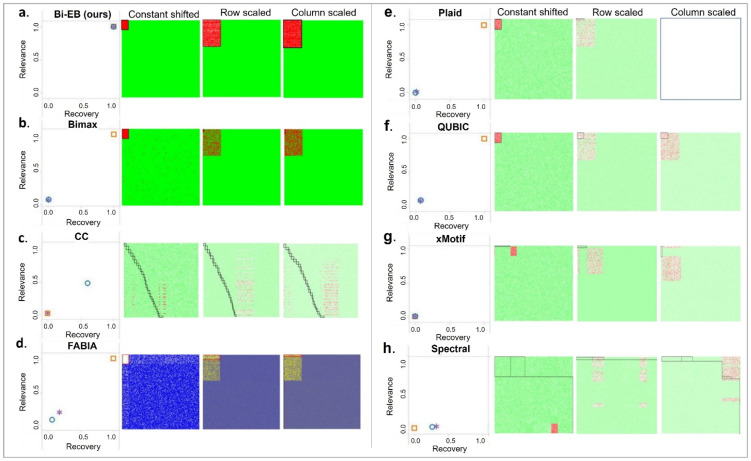
Bicluster model evaluation. Each group represents the average recovery versus relevance between the TRUE and predicted values in the biclustering algorithms: (**a**) BI-EB; (**b**) Bimax; (**c**) CC; (**d**) FABIA; (**e**) Plaid; (**f**) QUBIC; (**g**) xMotif; and (**h**) spectral relevance to constant-shifted, row-scaled, and column-scaled biclusters.

**Figure 4 genes-13-01982-f004:**
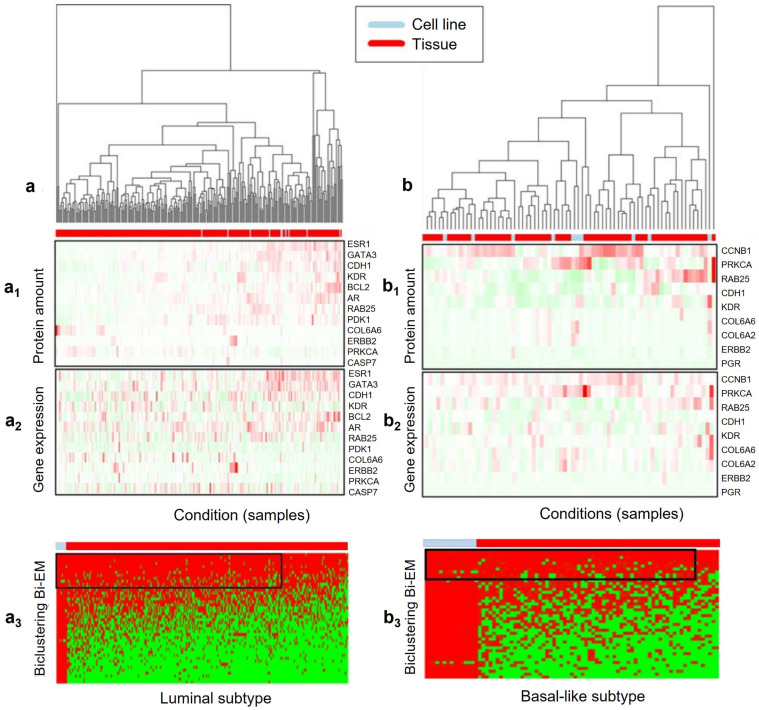
Heat map of membership assignment and extracted biclusters in (**a**) luminal and (**b**) basal-like subtypes. (**a1**,**a2**) are expression clustering under different conditions to Luminal subtype samples in protein expression data (**a1**) and mRNA expression data (**a2**). (**a3**) is the bi-cluster of ratios of protein amount in (**a1**) verse mRNA gene expression in (**a2**) by Bi-EM algorithm. (**b1**,**b2**) are expression clustering under different conditions to Basal-like subtype samples in protein expression data (**b1**) and mRNA gene expression data (**b2**). (**b3**) is the bi-cluster of ratios of protein amount in (**b1**) verse mRNA gene expression in (**b2**) by Bi-EM algorithm. Red shows the higher probability of belonging to a bicluster and green shows the lower probability of belonging to a bicluster in (**a3**,**b3**).

**Figure 5 genes-13-01982-f005:**
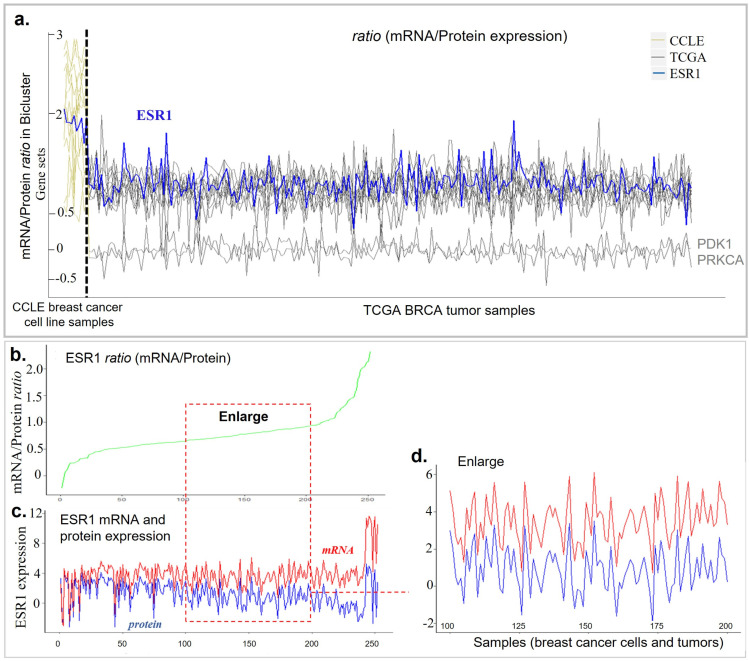
(**a**) Changes in the mRNA–protein ratio level of all genes across samples in the luminal A/B bicluster in breast cancer. The gray line is the ratio level of the gene in the cancer cell line (CCLE), the yellow line is the ratio level of the gene in tumor TCGA, and the blue line is the ratio level of gene ESR1. (**b**) The mRNA-protein ratio level of ESR1 across samples in the bicluster. Samples are sorted by ratio measurement. (**c**) The expression level of gene ESR1 (red) and protein ER (blue) across all samples in the luminal bicluster. Samples keep the same order as in ratio in (**b**). (**d**) The mRNA–protein ratio level of ESR1 across 100 samples in the bicluster.

**Figure 6 genes-13-01982-f006:**
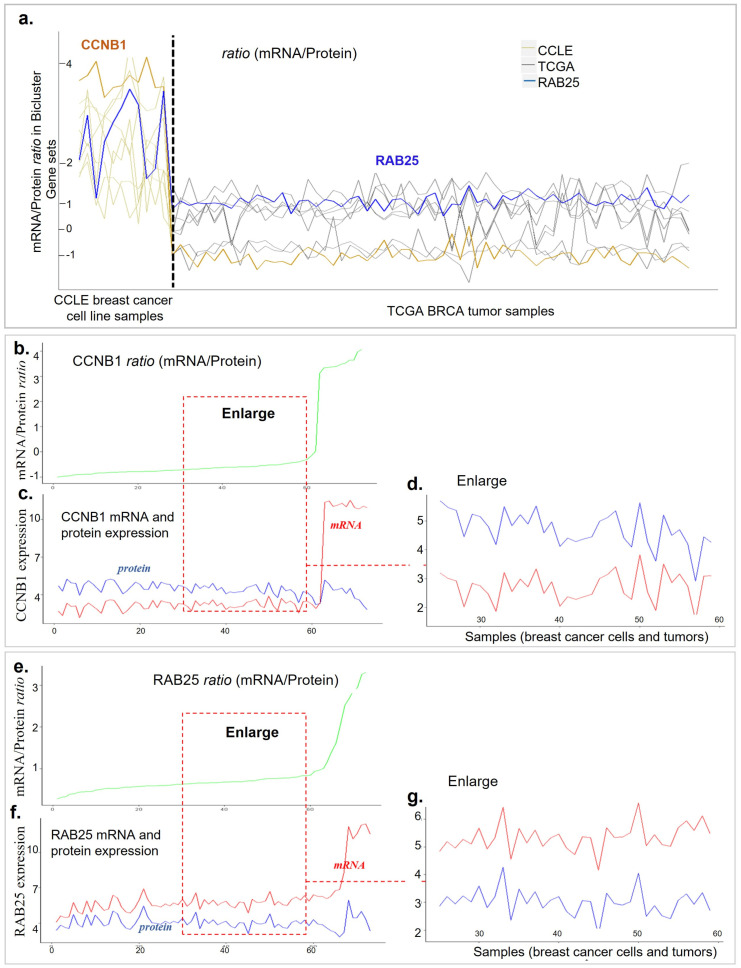
(**a**) Changes in the mRNA–protein ratio level of all genes across samples in the basal-like bicluster in breast cancer. The blue line is the ratio level of RAB25 and the gold line is CCNB1. (**b**) The mRNA–protein ratio level of CCNB1 across samples in the bicluster. Samples are sorted by ratio measurement. (**c**) The expression level of gene CCNB1 (red) and protein (blue) across all samples in the basal-like bicluster. Samples keep the same order as in ratio in (**b**). (**d**) The mRNA–protein ratio level of CCNB1 across 34 samples in the bicluster. (**e**) The mRNA–protein ratio level of RAB25 across samples in the bicluster. Samples are sorted by ratio measurement. (**f**) The expression level of gene RAB25 (red) and protein (blue) across all samples in the basal-like bicluster. Samples keep the same order as in ratio in (**e**). (**g**) The mRNA–protein ratio level of RAB25 across 34 samples in the bicluster.

**Table 1 genes-13-01982-t001:** Flow-chart for the biclustering Bi-EB algorithm.

**Inputs: Observed Data Matrix** $Y=\left\{y_{ij}\right\}$
**Data preprocessing:** Remove the data batch effect, normalize the data, and input the missing incomplete data
**Fitting the empirical Bayes biclustering model using the EM algorithm:** Initial values of $\theta^{\left(0\right)}=\left\{\theta_{1}^{\left(0\right)},\theta_{2}^{\left(0\right)}\right\}$ and $\;p^{\left(0\right)}=\left\{p_{1}^{\left(0\right)},p_{2}^{\left(0\right)}\right\}$. For iteration *t*∈1, 2, …, *N* do Evaluate probabilities belonging to a bicluster $P_{z}^{\left(t\right)}\leftarrow \log Pr\left(z|y_{ij};\;\theta^{\left(t\right)}\right)$ (*E*-step) $\;\;\;\;\theta^{\left(t+1\right)}\leftarrow argmax_{\theta }\left(l_{Ac}\left(Y,z;\theta^{\left(t\right)}\right)\right)$ (*M*-step) then return ${\hat{\theta }}^{\left(t+1\right)}$ Until $l_{Ac}\left(Y,z;\theta^{\left(t+1\right)}\right)\;-l_{Ac}\left(Y,z;\theta^{\left(t\right)}\right)<\epsilon$
**Searching for specific bicluster:** Set seed (such as druggable target gene) for initial searching and parameters ***A****c* and$\;p_{ave}$; Sort gene set *i* and sample set *j* in decreasing order by number of **1**s and **0**s; Arrange bicluster based on ‘***A****c**’* and ‘$p_{ave}$*’*.
**Output:****all** biclusters, *B_1,_ B_2,_* …*Bi*.

**Table 2 genes-13-01982-t002:** Biclustering algorithms and their parameters setting.

Algorithm Name	Year	Parameters	Available Software	Reference
**Cheng and Church**	2000	The optimization threshold (δ) and the number of biclusters n	R	[2]
**xMOTIFs**	2003	The optimization threshold, the size of the bicluster threshold, the number of gene thresholds per iteration, and the number of genes in the initial bicluster	R	[24]
**BiMAX**	2006	The size of biclusters n	R, Java	[25]
**Plaid**	2002	The number of biclusters, the number of iterations, amd the probability of in/excluding a gene during the clustering process	R	[4]
**Spectral**	2003	The number of biclusters, the optimization threshold, and the size of the bicluster threshold	R	[26]
**FABIA**	2010	The number of biclusters, the optimization threshold, the number of iterations, and the model-based parameter	R	[27]
**QUBIC**	2009	The number of biclusters, the optimization threshold, and the overlap threshold for obtained biclusters	R, C	[28]

## Data Availability

All data generated or analyzed during this study are included in this published article and its Appendix A. We share all Bi-EB R source codes and Bi-EB R packages, along with a tutorial and additional demo to guide users through our first example (involving CCLE and TCGA breast cancer gene expression data and protein expression data, accessed on 1 March 2022), at website https://github.com/lijcheng12/Bi-EB/, accessed on 21 September 2022.

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
