# Peer review of "Bi-EB: Empirical Bayesian Biclustering for Multi-Omics Data Integration Pattern Identification among Species"

_genes, 2022, doi:10.3390/genes13111982_

Round 1
Reviewer 1 Report
The authors aimed to propose an Empirical Bayesian Bi-clustering algorithm (Bi-EB) to identify translational gene sets shared between cancer cell lines and primary tumors based on mRNA and proteomics data or copy number variation (CNV) and mRNA data. An Expected-Maximum (EM) optimal algorithm was developed to conduct estimation and inference in the bi-cluster analyses.
The data were important, because their bicluster searching starts from a seed, such as a druggable target gene, and it detects interesting gene modules shared between cancer cell lines vs patient tumor samples. Using Bi-EB, gene modules of mRNA and proteomics are explored in two breast cancer sub-types, Luminal A/B and basal-like.
However, limitations of the study needs to be added.
Reviewer 2 Report
The authors proposed an empirical Bayesian bi-clustering approach for cross pattern identification across species. The paper is not well written. The usefulness of the proposed method is uncertain to me.
(1) There are lots of typos throughout the paper: “we present the of Bi-EB model” (line 139, page 3); “…and tissue samples, and. Here, …” (line 151, page 4); “In Figure 2e, …” (line 194, page 5), but there is no Figure 2e in the manuscript; “Red line is ratio…” (line 203, page 6), but there is no red line in Figure 2a.; In Figure 4, “Binarize by cut-off Ac”, but the cut-off is denoted by “c” (line 431, page 15); and etc.
(2) The notations are inconsistent (and there are many typos) in the method part, which makes the methodology hard to understand. For example, the bicluster is denoted by B (line 372, page 11), but it is denoted by B in bold on line 379. Do B and B in bold stand for the same bicluster? In Figure 4b, the bicluster is denoted by Bi. What are their differences? And the notations for matrix U are also different (line 446 vs line 448, page 16). On the other hand, how many biclusters can the proposed method identify? In the method part, the authors sometimes use “biclusters” (e.g., Figure 4D) and sometimes use “bicluster” (e.g., Table 1).
(3) It seems to me that the proposed method can only identify one bicluster, while most existing can identify multiple biclusters. It is true? If so, the simulation may not be able to support the conclusion that “it outperforms other 7 bi-clustering methods” (line 28, page 1), since only one bicluster is simulated. If not, how does the proposed method identify multiple biclusters? Could the authors give more details in the method part?
(4) On lines 450-451, page 16, the authors say “The final bicluster B is defined by a sub-matrix of U in which at least ???? of the elements equal to 1”. However, it is still unclear to me how to choose the sub-matrix? Could the authors provide some rigorous formulas?
(5) In the simulation, only one bicluster is simulated, and three indices (Jaccard coefficient, recovery and relevance) are used to compare the performances among methods. However, when there is only one bicluster, it seems to me that all three indices are the same for the proposed method.
(6) The presentation and interpretation of the results (Figures 1-3) are unclear. For example,
(6.1) For Figure 1 a1-a2, the authors say “The RPPA expression pattern does not show any relation to its corresponding mRNA pattern in the bicluster…” (lines 176-178, page 4). However, in Figure 2c, the authors say “both protein and mRNA levels of ESR1…appear to be correlated”. Does it mean that Figure 1 a1-a2 and Figure 2c lead to opposite conclusions?
(6.2) Are Figures 1 a and b the results for protein data using a hierarchical clustering approach? Is there any implication from these results?
(6.3) In Figure 1 a3-b3, what does the black frame represent?
(6.4) In Figure 2a, what does each line represent? Do different lines correspond to different genes? Why does the trend of ESR1 in Figure 2a look so different from that in Figure 2b? Both of their y-axes represent mRNA/Protein ratio, and x-axes are for the samples. In Figure 2a, the ratio for ESR1 is generally larger than 2, but it is generally smaller than 2 in Figure 2b.
(6.5) On line 189, page5, “Figure 2a further shows that there are two genes (PDK1 and PRKCA) whose GE/PA ratios are lower than other genes in bicluster”. But I cannot find this result in Figure 2a.
(7) Could the authors discuss the drawbacks of the proposed method?
Reviewer 3 Report
The manuscript by Yazdanparast, et al., titled "Bi-EB: Empirical Bayesian Bi-clustering for multi-omics data integration pattern identification among species" describes development of an algorithm called "Empirical Bayesian Biclustering (Bi-EB)" algorithm for detecting the patterns shared from both integrated omics data and between species. The authors demonstrate the application of this algorithm on publicly available breast cancer related datasets including Cancer Genomics Atlas (TCGA) and Cancer Cell Line Encyclopedia (CCLE). The manuscript can be accepted for publication and the authors can try to attend some minor comments below:
1. A general comment: The authors use the term "gene expression (GE)" and "protein amount (GE/PA)." I assume the authors refer to mRNA expression as "gene expression." Is my assumption correct? If yes, I would recommend using the term mRNA expression to be more specific.
2. In line 264, the authors state that "mRNA level of 80% of genes in bicluster can predict their related protein level." Is this a generic statement across all datasets that they analyzed? Also, are there any unique features among this 80% of the genes as compard to the remaining 20%?
3. The authors could enlighten a bit on the gene sets identified by their algorithm on the various breast cancer subtypes to highlight the power of their method.
